# Seasonal Variations of the Relationship between Spectral Indexes and Land Surface Temperature Based on Local Climate Zones: A Study in Three Yangtze River Megacities

Yang Xiang [1,2], Yongqi Tang [1], Zhihua Wang [3], Chucai Peng [1], Chunbo Huang [4], Yuanyong Dian [1], Mingjun Teng [1] and Zhixiang Zhou [1,*]

1   College of Horticulture and Forestry Sciences/Hubei Engineering Technology Research Center for Forestry Information, Huazhong Agricultural University, Wuhan 430070, China
2   Department of Architecture, National University of Singapore, Singapore 117566, Singapore
3   School of Sustainable Engineering and the Built Environment, Arizona State University, Tempe, AZ 85287, USA
4   Research Center for Spatial Planning and Human-Environment System Simulation, School of Geography and Information Engineering, China University of Geosciences, Wuhan 430078, China
*   Correspondence: whzhouzx@mail.hzau.edu.cn

**Abstract:** Urban heat islands are representative problems in urban environments. The impact of spectral indexes on land-surface temperature (LST) under different urban forms, climates, and functions is not fully understood. Local climate zones (LCZs) are used to characterize heterogeneous cities. In this study, we quantified the contribution of three cities to high-temperature zones and surface urban heat island intensity (SUHII) across LCZs and seasons, used Welch and Games–Howell tests to analyze the difference in LST, then described the spatial pattern characteristics of LST, and used a geographically weighted regression model to analyze the relationship between spectral indexes and LST. The results showed that compact midrise, compact low-rise (LCZ 3), large low-rise (LCZ 8), heavy industry (LCZ 10), and bare rock or paved (LCZ E) contributed greatly to high-temperature zones and had strong SUHII. There were 92–98% significant differences between different LCZs. The spatial aggregation of LST gradually weakened with a decrease in temperature. The modified normalized difference water index (MNDWI) in most LCZs of all seasons for Wuhan could reduce LST well, while MNDWI only had cooling effects in winter for Nanjing and Shanghai. Normalized difference vegetation index (NDVI) in most LCZs performed a cooling role during summer and transition seasons (spring and autumn), while it showed a warming effect in winter. The cooling effect of NDVI in open building types was stronger than that of compact building types, while the cooling effect of MNDWI was better in compact building types than in open building types. With the increase of normalized difference built-up index (NDBI), all LCZs showed warming effects, and the magnitude of LST increase varied in different cities and seasons. These results contribute further insight into thermal environment in heterogeneous urban areas.

**Keywords:** spectral indexes; local climate zones; surface urban heat island intensity; urban heat islands; land surface temperature; geographically weighted regression

## 1. Introduction

Urbanization is an important trend in global development [1]. More than half of the world's population lives in cities, which is expected to rise to 66% by 2050 with further urbanization [2]. The process of urbanization, especially in China, has accelerated since the late 1970s [3]. Rapid urbanization has caused a widespread land use conversion from natural vegetation and agricultural landscapes to impervious surfaces. This transformation is usually accompanied by dramatic changes in the energy exchange between the urban surface and the atmosphere, which has led to numerous ecological and environmental

problems [4]. Urban heat island (UHI) is a common environmental problem [5], defined as a phenomenon that urban cores are usually warmer than their rural surroundings [6]. Detrimental effects of UHI have been widely reported, and it has increased building cooling consumption [7], worsened air pollutants [8], seriously harmed human health, and even raised heat-related deaths of urban residents [9], altered species composition and abundance [10], and synergized with global climate change [11]. In cold climates, UHIs have some benefits, including reduced energy consumption, improved thermal comfort, and reduced cold-related mortality [12]. Therefore, it is extremely important to further understand UHI in highly heterogeneous urban environments, which can help urban planners and policymakers to propose adaptive measures.

Surface urban heat island (SUHI) driven by remote sensing has been greatly developed due to the advantages of easy access, wide spatial distribution, and short time interval [12–14]. Land surface temperature (LST) is used to characterize SUHI, and the relationships between LST and influencing factors are widely quantified [13]. These influencing factors include land cover, urban form (building density, building height, and sky view factor), socioeconomic indicators (nighttime lights, gross domestic product, and population), spectral indices, climate conditions (air temperature, precipitation, and humidity), anthropogenic emission of heat, and materials of impervious surfaces [13–18]. Different types of land cover significantly affect LST variation and are described by many spectral indices, which are easy to obtain from remote sensing information and contain rich land cover information [19]. Modified normalized difference water index (MNDWI) shows a negative correlation with LST [20]. Normalized difference vegetation index (NDVI) is used to describe vegetation status and greenness, and it has a significant negative correlation with LST [12]. Normalized-difference built-up index (NDBI) describes building characteristics, is an indicator of grey infrastructure, and has a significant positive correlation with LST [19]. Peng et al. (2018) indicated that the dominant factors affecting LST were NDBI in summer and NDVI in the transition season and winter, and these indices had more explanations than land cover indexes [19]. Xie et al. (2018) indicated that LST decreased by 1.4, 1.7, 1.3, and 1.8 °C for each 0.1 increase in NDVI for built-up areas, farmland, grassland, and forest, respectively [20]. Liu et al. (2022) indicated that every 0.1 unit of NDVI and NDMI would have cooling effects of 1.1 to 2.0 and 2.8 to 4.1 °C, respectively, while NDBI would result in a warming effect of 1.4 to 2.3 °C [21].

Although these previous studies investigated the relationships between spectral indices and LST [22–27], most of these studies analyzed their relationships on a whole-city or land cover basis, and the relationship between urban landscape categories and LST under different urban forms, climate, and functions were not adequately studied due to the high heterogeneity of urban areas. Therefore, it is very necessary to study the relationships between spectral indices and LST under different urban landscapes based on the classification framework of urban forms, climate, and function. Local climate zone (LCZ) provides a unified classification framework for urban form, climate, and function [28,29]. LCZ is defined as a relatively homogeneous area with structure, material, and human activity including 10 built types and 7 land cover types, and more classification details can be found in Stewart and Oke's (2012) study [30]. The LCZ framework incorporates different aspects of urban forms, such as sky view factor, aspect ratio, building surface fraction, height of rough elements, terrain roughness class, and significant differences of climate [31,32], representing diverse urban functions: the functions of LCZ A and B correlate with natural forest, tree cultivation, or urban park; LCZ D with natural grassland, agriculture, or urban park; LCZ E with natural desert or urban transportation, and so on. LCZ classification has become a new classification for characterizing urban landscapes, as it is coupled with key factors influencing LST [1]. Therefore, to explore the relationship between spectral indexes and LST under different urban forms, climates, and functions, this study intends to analyze the relationship between MNDWI, NDVI, NDBI, and LST in different LCZs. The main objectives of this study are to (1) investigate the contribution of different LCZs to high-temperature zones and the surface heat island intensity (SUHII)

characteristics of different LCZs; (2) determine whether there are significant differences in LST under different LCZs; (3) characterize the spatial pattern characteristics of LST in different cities and seasons; and (4) quantify the relationships between spectral indices and LST under different LCZs. This study is expected to help urban planners and policymakers to better formulate development policies to mitigate UHI by understanding the thermal environment under the urban forms, climate, and functions described by LCZs.

## 2. Materials and Methods

### 2.1. Study Area

We selected the central districts of three cities—Wuhan, Nanjing, and Shanghai—as study areas (Figure 1). These are major cities in the Yangtze River, i.e., the capital of Hubei province, the capital of Jiangsu province, and a municipality directly under the central government, China, respectively. These cities are among the top 10 municipalities and provincial capitals in China in terms of regional GDP and had a resident population of over 9 million people in 2020. All cities are similar in climate background (Chinese climate zoning class 2—Jiangbei district and subtropical monsoon climate and abundant rainfall) (Figure 1a). They have experienced serious UHI with the rapid development of urbanization and high population density [33,34].

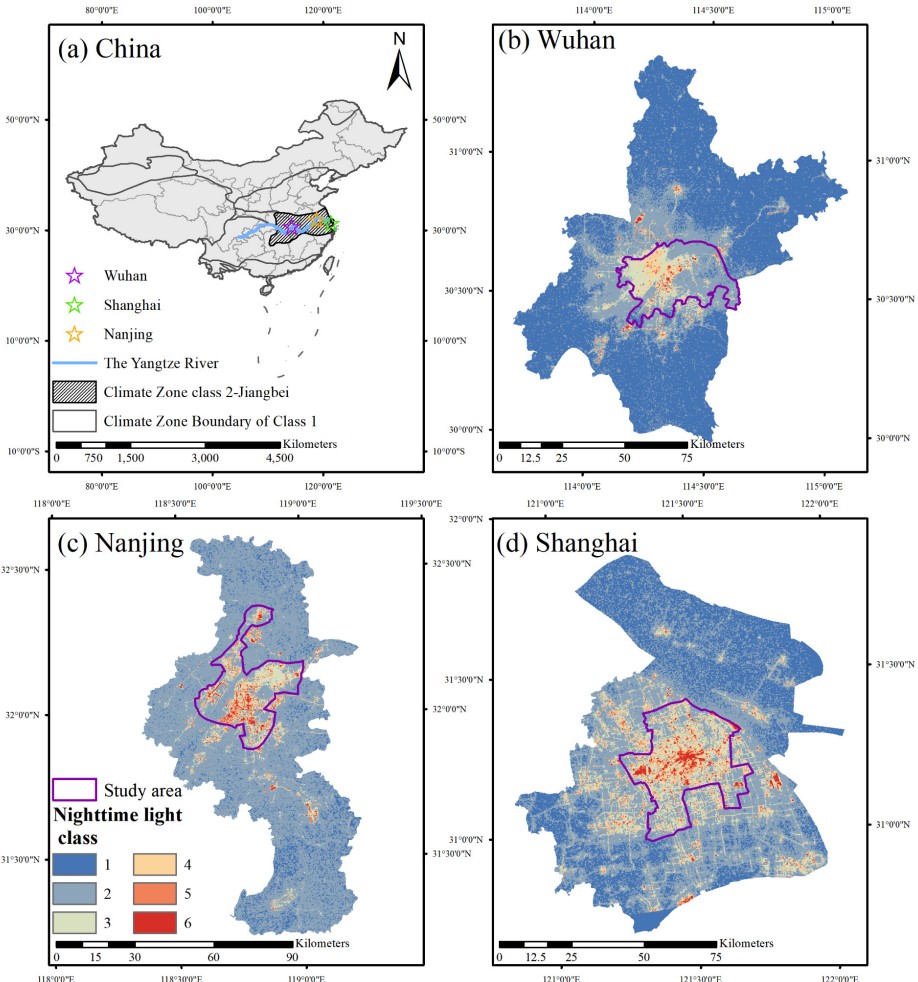

**Figure 1.** Location and study area: (**a**) is the location of the three cities in China, and (**b**–**d**) are Wuhan, Nanjing, and Shanghai respectively. The quantile classification method is used for night lighting intensities.

*2.2. Data Collection*

The Landsat 8 data (LANDSAT/LC08/C02/T1_L2) came from the Google Earth Engine (https://earthengine.google.com/ (accessed on 1 April 2022)) in which all bands have been calibrated radiometrically and atmospherically. LST came from Landsat8 "ST_B10" band, inverted by a single-channel algorithm. Landsat 8 with cloud coverage greater than 20 were rejected, and the areas covered by clouds in Landsat 8 were masked. Missing values were first filled using close times for neighboring years, and the rest were interpolated using a co-Kriging method. Due to the poor quality of summer data in Nanjing, the median synthesis method was used to characterize the summer data in 2020. More information about data, cloud cover, and path/row number of Landsat 8 can be found in Table A1.

In this study, summer includes June, July and August; transitional seasons include March, April, May, September, October, and November; winter includes December, January, and February of the following year. Transition seasons were calculated from the mean values of spring (March, April, and May) and autumn (September, October, and November) due to the similar patterns of UHI in spring and autumn [35,36].

*2.3. Methods*

2.3.1. Spectral Indexes

In this study, we selected MNDWI, NDVI, and NDBI as spectral indexes. The Equations (1)–(3) were calculated as follows:

$$MNDWI = \frac{\rho_{GREEN} - \rho_{SWIR1}}{\rho_{GREEN} + \rho_{SWIR1}}, \tag{1}$$

$$NDVI = \frac{\rho_{NIR} - \rho_{RED}}{\rho_{NIR} + \rho_{RED}}, \tag{2}$$

$$NDBI = \frac{\rho_{SWIR1} - \rho_{NIR}}{\rho_{SWIR1} + \rho_{NIR}}, \tag{3}$$

where $\rho_{NIR}$ is the near-infrared band, $\rho_{RED}$ is the red band, $\rho_{SWIR1}$ is the short-wave infrared band, and $\rho_{GREEN}$ is the green band.

2.3.2. Classification of Local Climate Zones

We used the LCZ Generator proposed by Demuzere (2021) to classify LCZs, which is based on the classical World Urban Database and Access Portal Tools (WUDAPT) workflow, using Landsat data and combining Sentinel-1, Sentinel-2, global forest canopy height, digital elevation model, and other earth observation data by the random forest algorithm [37]. LCZ Generator has more robust accuracy assessments during the production of LCZ and allows convenient quality control by allowing the user to revise training area, as the website integrates the WUDAPT process and only requires training data of each LCZ to the URL (https://lcz-generator.rub.de (accessed on 30 January 2022)) to generate classification results and quality assessment. More processes about LCZs can be found in Figure A1. In this study, the training data for the entire cities of Wuhan, Nanjing, and Shanghai in 2020 were created in Google Earth Pro, where 1997, 1443, and 1716 polygons were obtained for generating LCZs, respectively. Description of LCZs can be found in Table 1.

2.3.3. Contribution of High-Temperature Zone

Based on previous studies [38–40], we used the natural breakpoint method to classify the LST in different cities and seasons into four categories: high-temperature zone, sub-high-temperature zone, sub-low-temperature zone, and low-temperature zone. The contribution index (CI) was used to describe the thermal contribution of each LCZ to the high-temperature zone. The formula is as follows:

$$CI = \frac{S_{hi}}{S_i} / \frac{S_h}{S}, \tag{4}$$

**Table 1.** Definitions of local climate zones.

| Built Types | | Land Cover Types | |
| --- | --- | --- | --- |
| Class | Description | Class | Description |
| LCZ 1 | Compact high-rise | LCZ A | Dense trees |
| LCZ 2 | Compact mid-rise | LCZ B | Scattered trees |
| LCZ 3 | Compact low-rise | LCZ C | Bush, scrub |
| LCZ 4 | Open high-rise | LCZ D | Low plants |
| LCZ 5 | Open mid-rise | LCZ E | Bare rock or paved |
| LCZ 6 | Open low-rise | LCZ F | Bare soil or sand |
| LCZ 8 | Large low-rise | LCZ G | Water |
| LCZ 9 | Sparsely built | | |
| LCZ 10 | Heavy industry | | |

where i is the index of LCZ type, $S_{hi}$ is the area of the high-temperature zone in a type i LCZ, $S_i$ is the area of a type i LCZ, and S is the total area of the central district. CI > 1 indicates that the distribution frequency of the high-temperature zone under the LCZ type is higher than that of the central district and provides a positive contribution to the thermal environment of the central district; CI < 1 indicates that the LCZ type provides a negative contribution to the thermal environment of the central district.

2.3.4. Analysis of Surface Urban Heat Islands Intensity

The SUHII is defined as the difference in LST between a particular type of LCZ and LCZ D (low vegetation), as LCZ D could characterize non-urban landscapes and reflect spatial variation [30,32,41]. To investigate the seasonal variation in SUHII for each LCZ, SUHII was calculated for the three cities in different seasons. SUHII was calculated as follows:

$$SUHII_{LCZ\ X} = LST_{LCZ\ X} - LST_{LCZ\ D}, \tag{5}$$

where LCZ X is type X of LCZ, $SUHII_{LCZ\ X}$ is SUHII of LCZ X, $LST_{LCZ\ X}$ is the mean of LST in LCZ X, and $LST_{LCZ\ D}$ is the mean of LST in LCZ D.

2.3.5. Analysis of Variance and Post Hoc Test

Analysis of variance (ANOVA) was used to compare whether there were significant differences in LST between LCZs. ANOVA requires assumptions of normality and homogeneity of variance [42]. Shapiro–Wilk and Kolmogorov–Smirnov tests were used to test for normality of LST in each type of LCZ, and the Levene test was used to test for homogeneity of variance. Tables A2 and A3 showed that both normality and homogeneity of variance did not pass significance. Therefore, the Welch's ANOVA and Games–Howell post hoc test were used to determine significant differences in LST of LCZ. These are, respectively, the classical and alternative method when the prerequisites of normality and homogeneity of variance are not met [43].

2.3.6. Spatial Pattern Characteristics of Land Surface Temperature

In this study, the global and local Moran index were used to quantify the spatial autocorrelation characteristics of LST in different seasons. The global Moran index was calculated as shown:

$$I = \frac{\sum_{i=1}^{n}\sum_{j=1}^{n} w_{ij}(x_i - \bar{x})(x_j - \bar{x})}{\sigma^2 \sum_{i=1}^{n}\sum_{j=1}^{n} w_{ij}}, \tag{6}$$

$$\sigma = \frac{1}{n}\sum_{i=1}^{n}(x_i - \bar{x})^2, \tag{7}$$

where n represents the number of grids in the study area; $x_i$ and $x_j$ represent the LST of grids i and j, respectively; $\bar{x}$ is the average of all grids in the study area; $w_{ij}$ is the spatial weight matrix of grids i and j; and $\sigma$ is the standard deviation of all grids in the study area.

The local Moran index was calculated as shown:

$$I_L = \frac{\sum_{i=1}^{n}\sum_{j=1}^{n} w_{ij}(x_i - \bar{x})(x_J - \bar{x})}{\sigma^2},\tag{8}$$

Moran indexes ranged from [−1, 1], where if it is less than 0, there is a negative correlation; greater than 0 means a positive correlation; and equal to 0 means no correlation. The results of local Moran index can be classified into four types: high–high, low–low, high–low, and low–high. The high–high or low–low type refers to the aggregation characteristics of high or low LST, while the high–low or low–high type refers to the aggregation characteristics of high or low LST surrounded by low or high surface temperatures.

### 2.3.7. Geographically Weighted Regression

Geographically weighted regression (GWR) is an improvement on ordinary least squares, as its regression coefficients are not global but vary with space. It is a local regression and can be used to explore the spatial non-stationarity of influence factors [44]. GWR is calculated as follows:

$$y_i = \beta_0(u_i, v_i) + \sum_k \beta_k(u_i, v_i)x_{ik} + \varepsilon_i,\tag{9}$$

where $y_i$ is the LST of grid i, $(u_i, v_i)$ is the spatial coordinate of grid i, $\beta_0(u_i, v_i)$ is the intercept of grid i, k is the number of explanatory variables (MNDWI, NDVI, and NDBI) and equals 1 due to collinearity problems in these explanatory variables, $\beta_k(u_i, v_i)$ is the regression coefficient of grid i at $(u_i, v_i)$, and $x_{ik}$ is the value of the kth MNDWI, NDVI, or NDBI at grid i; $\varepsilon_i$ is the residual of grid i.

## 3. Results

### 3.1. Statistical Characteristics of Spectral Indexes and LST

The statistical characteristics of spectral indexes and LST are described in Table 2. There were seasonal variations in MNDWI, NDVI, NDBI, and LST. The highest LST in summer, transition season, and winter was in Shanghai; the lowest LST values for each season were in Nanjing, Wuhan, and Wuhan, respectively. Wuhan had the highest MNDWI values in all seasons, and the lowest were in Shanghai, Shanghai, and Nanjing, respectively. Shanghai had the highest NDBI in different seasons, but the lowest was in Wuhan. The highest NDVI in different seasons was in Nanjing, and the lowest values were in Nanjing, Shanghai, and Shanghai, respectively. The dispersion of LST, MNDWI, NDVI, and NDBI between seasons was the largest in Wuhan, especially from summer to the transition season, which might be due to the rich LCZ type of land cover.

**Table 2.** Statistical characteristics of MNDWI, NDVI, NDBI, and LST.

| City | Season | LST (°C) | MNDWI | NDBI | NDVI |
|---|---|---|---|---|---|
| | Summer | 44.00 ± 7.02 | −0.14 ± 0.45 | −0.18 ± 0.19 | 0.28 ± 0.41 |
| Wuhan | Transition season | 27.57 ± 4.25 | −0.10 ± 0.34 | −0.16 ± 0.13 | 0.25 ± 0.29 |
| | Winter | 11.65 ± 1.89 | 0.001 ± 0.31 | −0.11 ± 0.14 | 0.12 ± 0.21 |
| | Summer | 39.20 ± 4.16 | −0.25 ± 0.25 | −0.18 ± 0.13 | 0.42 ± 0.25 |
| Nanjing | Transition season | 27.93 ± 2.84 | −0.20 ± 0.24 | −0.14 ± 0.11 | 0.33 ± 0.22 |
| | Winter | 15.63 ± 2.40 | −0.19 ± 0.26 | −0.05 ± 0.11 | 0.21 ± 0.20 |
| | Summer | 45.25 ± 3.46 | −0.26 ± 0.19 | −0.14 ± 0.13 | 0.38 ± 0.24 |
| Shanghai | Transition season | 37.15 ± 3.28 | −0.25 ± 0.16 | −0.12 ± 0.13 | 0.34 ± 0.22 |
| | Winter | 15.75 ± 2.22 | −0.17 ± 0.17 | −0.07 ± 0.10 | 0.22 ± 0.16 |

The magnitude of decrease in LST, MNDWI, NDBI, and NDVI varied among cities between seasons. These indicators showed the greatest decrease from the transition season to winter compared to the decrease from summer to the transition season. The reductions in LST and MNDWI were ranked from Wuhan (57.74% and 101.00%), Shanghai (57.60% and

32.00%), and Nanjing (44.04% and 5.00%), respectively, while the decreases from summer to the transition season were from Wuhan (37.34% and 28.57%), Nanjing (28.75% and 20.00%), and Shanghai (17.90% and 3.85%), respectively. Unlike LST and MNDWI, the declines in NDBI from the transition season to winter were in Nanjing (64.29%), Shanghai (41.67%), and Wuhan (31.25%); the reductions from summer to transition season were ranked by Nanjing (22.22%), Shanghai (14.29%), and Wuhan (11.11%), respectively. The NDVI in Wuhan, Nanjing, and Shanghai decreased by 52.00%, 36.36%, and 35.29% from the transition season to winter, respectively, but decreased by 10.71%, 21.43%, and 10.53% from summer to the transition season, respectively. Compared with Shanghai and Nanjing, Wuhan has abundant agricultural land within the main urban area, and the harvesting of crops leads to an increase in LST and large fluctuations in MNDWI and NDVI.

### 3.2. Classification of Local Climate Zones

The classification of LCZs was carried out based on a random forest algorithm, where 70% of the polygon data were used for training and 30% for testing. Four metrics were used to describe the classification accuracy: overall accuracy (OA), overall accuracy for the urban LCZ classes only ($OA_u$), overall accuracy of the built vs. natural LCZ classes only ($OA_{bu}$), and weighted accuracy ($OA_w$). OA and $OA_u$ can reflect the percentage of cells that are correctly classified for all LCZs and for built type LCZs, respectively, while $OA_{bu}$ is the overall accuracy for building vs. land cover LCZs, and $OA_w$ is calculated by weighting the confusion matrix. The OA, $OA_u$, $OA_{bu}$, and $OA_w$ were 0.86, 0.76, 0.97, and 0.97 for Wuhan; 0.81, 0.70, 0.97, and 0.96 for Nanjing; and 0.79, 0.74, 0.94, and 0.95 for Shanghai, respectively. These results have better accuracy than those in previous studies [45,46] due to the support of a large number of training samples and could reflect LCZs more accurately. The spatial distribution and the percentage of each type of LCZ are shown in Figure 2. The percentage of building types exceeded 50% in all cities, ranked from highest to lowest in Shanghai, Nanjing, and Wuhan, respectively. All cities had been dominated by open LCZ 4 and LCZ 5, which were more than one-third. In all cities, land cover types were dominated by LCZ D and LCZ G, and Wuhan had the highest percentage of these land cover types, with nearly 50%. Built types of high density were arranged in LCZ G (Yangtze and Huangpu rivers).

### 3.3. Contribution of Local Climatic Zones to High-Temperature Zones

Figure 3 depicts the contribution of each LCZ in the three cities to their central district, with heterogeneity and seasonal variations. In summer, CIs of LCZ 1, 2, 3, 8, 10, and E were greater than other LCZs in the three cities, indicating that these LCZ types contributed positively to high-temperature zones in the central district, increasing the thermal risk. LCZ 4, 5, and 6 showed differences in CIs across cities, which may be related to differences in land composition among LCZ types. In contrast, CIs of LCZ 9, A, B, C, D, F, and G were less than 1, which means that these LCZs contributed negatively to high-temperature zones, and increasing these LCZs could reduce LST well. In the transition seasons, LCZ 2, 3, 5, 8, 10, and E contributed positively, and LCZ 9, A, B, D, and G contributed negatively; in winter, LCZs with positive contributions to the high-temperature zones gradually decreased and only included LCZ 2, 8, 10, and F. Concurrently, LCZs with negative contribution to high-temperature zones were LCZ 5, 6, 9, A, B, D, and G.

### 3.4. Surface Urban Heat Island Intensity

The results of SUHII are shown in Figure 4. They exhibits significant seasonal variations due to differences in radiation intensity and sunshine duration between seasons, with the strongest SUHII in summer, followed by the transitional seasons, and the lowest SUHII in winter. SUHII in summer was most prominent and significant, which is consistent with previous studies [18]. For the magnitude of SUHII, although differences appear across cities, LCZ 2, 3, 8, 10, and E had the higher SUHIIs in all cities and seasons, while the SUHII of LCZ A and G was lower. The SUHII of LCZ 1 was less than that of LCZ 4 in most cases, probably due to the mutual shading between buildings in LCZ 1 blocking part of the

radiation. SUHIIs of LCZ 2 and LCZ 3 were both stronger than those of LCZ 5 and LCZ 6. It can be found that the magnitudes of SUHII in Wuhan were the largest in summer and the transition season at 21.41 and 12.31 °C, respectively, which might be due to the abundant land cover type in the central district of Wuhan, especially LCZ D, accounting for more than one-fourth. The magnitude of SUHII in winter was not as large as in other seasons.

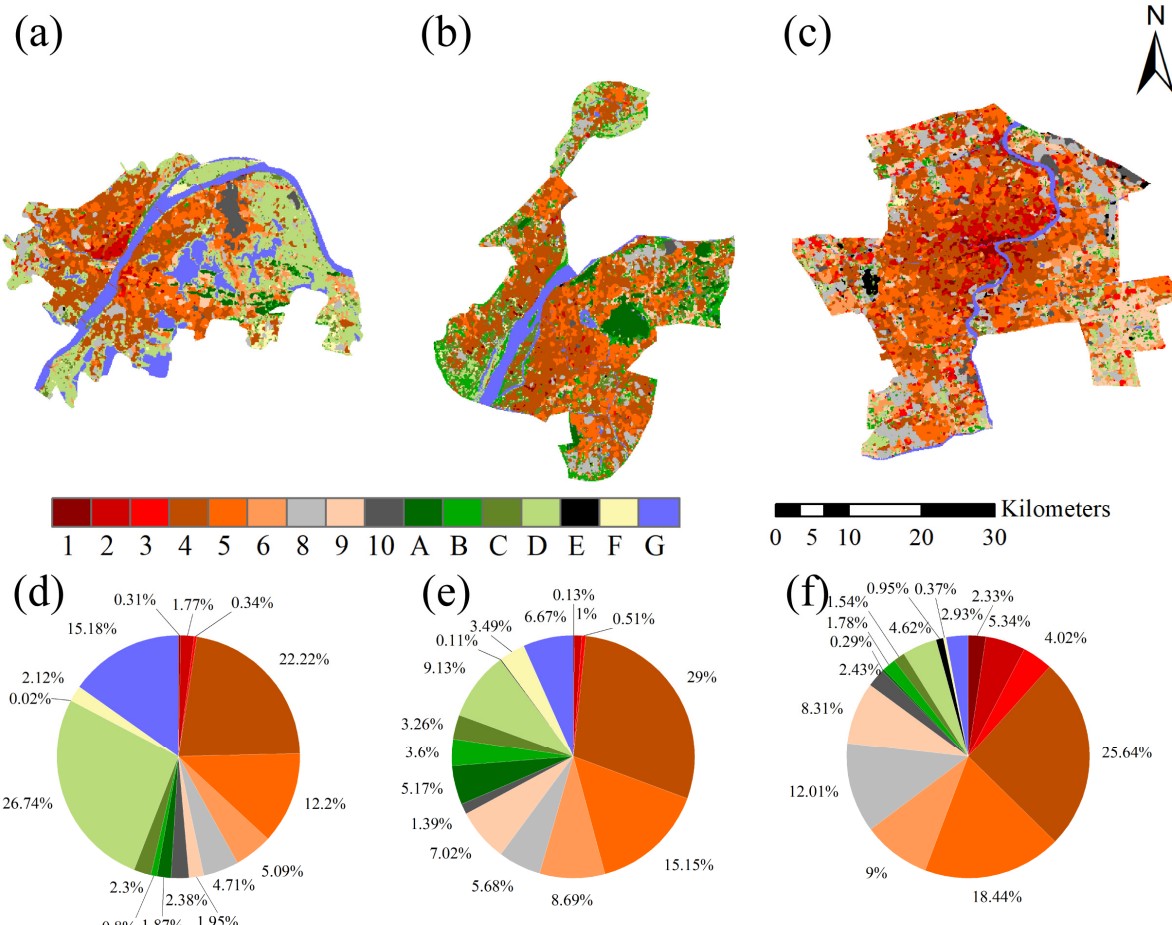

**Figure 2.** Spatial distribution and pie charts of LCZ in three cities: (**a**–**c**) are spatial distributions of local climate zones at Wuhan, Nanjing, and Shanghai; (**d**–**f**) show pie chart of local climate zones at Wuhan, Nanjing, and Shanghai, respectively.

### 3.5. Analysis of Variance and Post Hoc Comparisons

The results of Welch's test showed significant differences ($p < 0.05$) between LCZs in all cities and seasons (Table 3). The Games–Howell test was applied to test whether there was a significant difference between any two LCZs. The results showed significant differences in more than 90% of the two-way comparisons; specifically, the percentages in Wuhan with significant differences in summer, transition season, and winter were 95%, 92%, and 94%, respectively; in Nanjing, 98%, 97%, and 93%; and in Shanghai, 95%, 98%, and 93%, respectively (Figure 5). In general, the percentage with significant differences was lower in winter compared to other seasons.

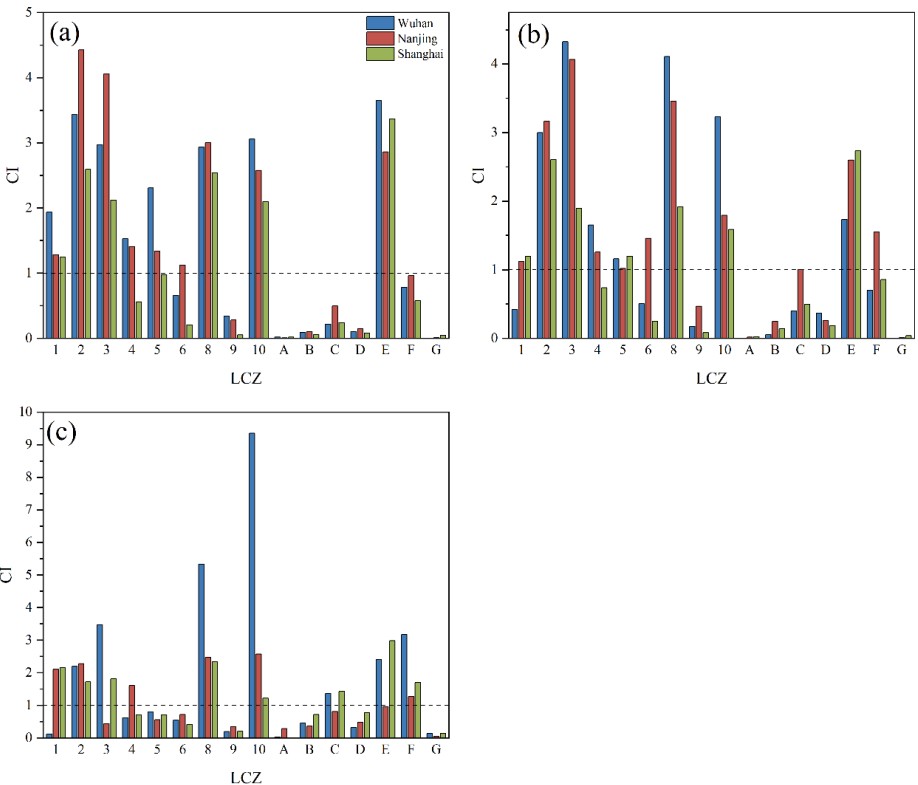

**Figure 3.** Comparison of results of contribution index to high-temperature zones in three cities: (**a**) summer, (**b**) the transitional seasons (spring and fall), and (**c**) winter in 2020.

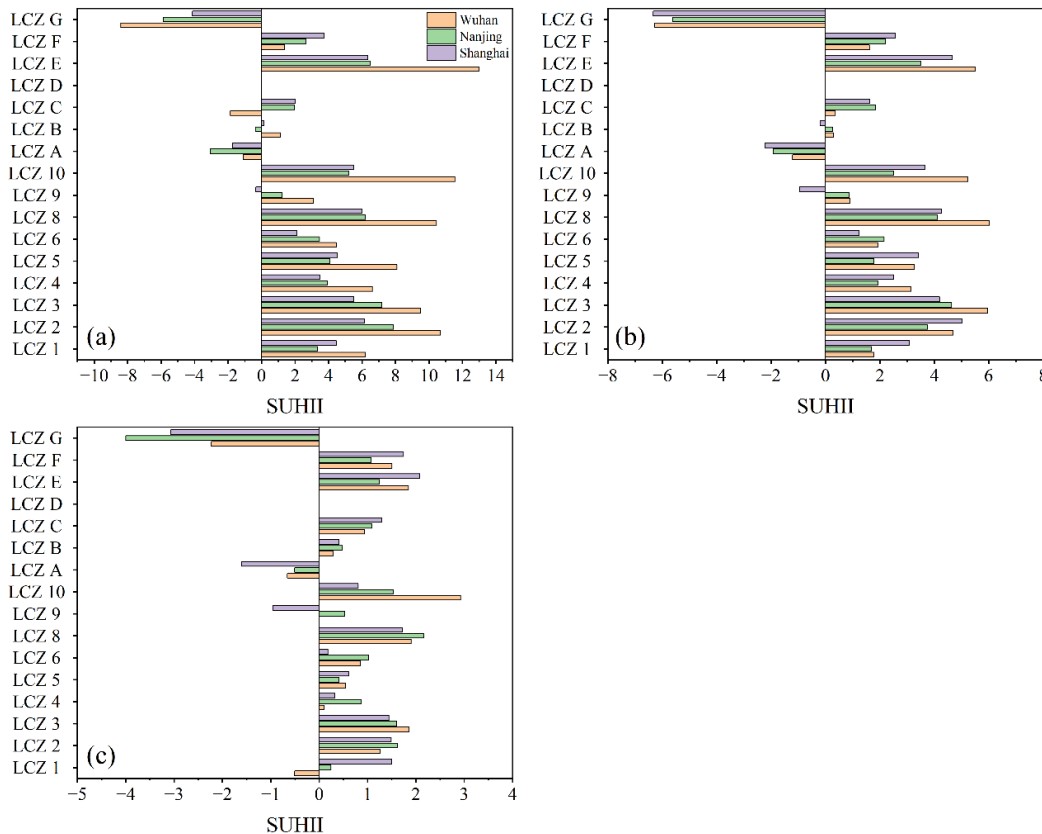

**Figure 4.** Results for surface urban heat-island intensity in three cities and seasons: (**a**) summer, (**b**) the transitional seasons (spring and fall), and (**c**) winter in 2020.

**Table 3.** Results of Welch tests in different local climate zones.

| City | Season | Statistic | df1 | df2 | Significance |
|---|---|---|---|---|---|
| Wuhan | Summer | 19,310.390 | 15 | 1028.742 | 0.000 |
| | Transition season | 19,288.741 | 15 | 1028.556 | 0.000 |
| | Winter | 2747.871 | 15 | 1030.775 | 0.000 |
| Nanjing | Summer | 8016.538 | 15 | 1741.385 | 0.000 |
| | Transition season | 6199.907 | 15 | 1741.496 | 0.000 |
| | Winter | 3365.614 | 15 | 1743.017 | 0.000 |
| Shanghai | Summer | 5808.144 | 15 | 9367.378 | 0.000 |
| | Transition season | 4903.839 | 15 | 9364.059 | 0.000 |
| | Winter | 1425.392 | 15 | 9378.346 | 0.000 |

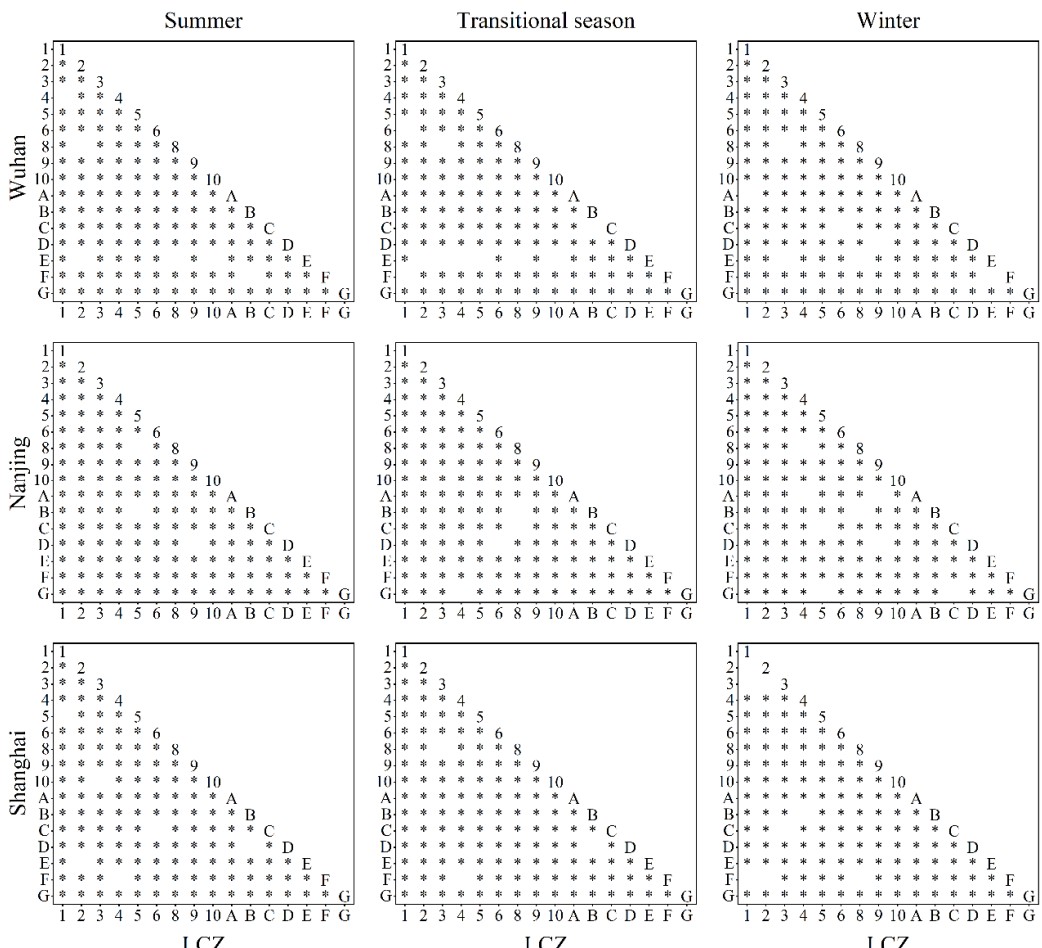

**Figure 5.** Results of post hoc comparisons between LCZs in different cities and seasons, and * represents differences between LCZ ($p < 0.05$).

### 3.6. Spatial Pattern Characteristics of Land Surface Temperature

In this study, global and local Moran indices were used to indicate the spatial aggregation characteristics of LST in different cities and seasons. The results in Table 4 demonstrated that LST had different spatial agglomeration characteristics in different cities and seasons, and the strength of these spatial agglomeration characteristics varied with temperature, weakening gradually from summer through the transition season to winter. Wuhan had the strongest spatial agglomeration characteristics, with Moran indexes of 0.95, 0.93, and 0.86 in the summer, transition seasons, and winter, respectively.

**Table 4.** Result of global Moran I in different seasons.

| City | Season | Moran's Index | z-Score | *p*-Value |
|---|---|---|---|---|
| Wuhan | Summer | 0.95 | 574.47 | <0.001 |
| | Transition season | 0.93 | 564.06 | <0.001 |
| | Winter | 0.86 | 520.92 | <0.001 |
| Nanjing | Summer | 0.92 | 386.88 | <0.001 |
| | Transition season | 0.91 | 383.73 | <0.001 |
| | Winter | 0.75 | 317.32 | <0.001 |
| Shanghai | Summer | 0.87 | 416.87 | <0.001 |
| | Transition season | 0.87 | 415.65 | <0.001 |
| | Winter | 0.82 | 389.50 | <0.001 |

The spatial agglomeration characteristics of the three cities showed a similar pattern across different seasons (Figure 6). Specifically, the distribution pattern of the "high–high" type in Wuhan was similar in different seasons, mainly concentrated in the western, southwestern, and central-eastern parts of the central district, and the distribution was more discrete in winter; the "low–low" type was distributed in the Yangtze River and in large lakes in the summer and transition season and in large lakes in winter. Moran indexes for Nanjing and Shanghai had a similar distribution pattern, in which the "high–high" type was discrete in the central districts of Nanjing and Shanghai; the "low–low" type in Nanjing was mainly distributed in the Yangtze River and in large lakes in the central-eastern part of the central district in the summer and transition seasons and in the central-eastern part of central district in winter; the "low–low" type in Nanjing was mainly distributed in the Yangtze River and the large green space in the central-eastern part of central district during the summer and transition seasons, while the significance of the large green space in the central-eastern part of the central district in Nanjing was greatly reduced in winter, probably due to the insulation function of green areas in winter [25]; the "low–low" type in Shanghai was mainly distributed in the Huangpu River and the southeast corner, and the aggregation characteristics were more obvious in winter.

*3.7. The Relationship between Spectral Indexes and LST*

Due to the strong collinearity between MNDWI, NDBI, and NDVI, we quantified the relationships between MNDWI, NDBI, NDVI, and LST, respectively. Table 5 describes the fit goodness of MNDWI, NDBI, and NDVI for LST in GWR. In general, the explanatory effects in summer and transition seasons were stronger than those in winter, and NDBI had the highest explanation for LST. We excluded regions where the standard residuals were not in [−2.5, 2.5], as these coefficients were unreliable [47,48].

Table 6 shows the change in LST with each 0.1 unit increase in MNDWI, NDVI, and NDBI. MNDWI is a indicator to describe the composition of a water body, which has high heat capacity and low thermal conductivity, and is able to reduce LST through evaporation. The change of LST showed seasonal variation with an increase of 0.1 unit of MNDWI. In general, the cooling effect of MNDWI decreased with the decrease of temperature, and the cooling amplitude decreased to varying degrees in building types. MNDWI could reduce LST well in most LCZs of all seasons for Wuhan, but for Nanjing and Shanghai, only MNDWI in winter had cooling effects in all LCZs, and the number of LCZs with cooling effects of MNDWI in the transition season was more than in summer. Variations in the cooling effects of MNDWI between cities might be due to differences between the number, size, and distribution of water bodies [49]. In the study, more cooling effects from MNDWI were found in Wuhan than in Nanjing and Shanghai, as Wuhan has abundant water resources with wide distribution and large numbers compared to Nanjing and Shanghai. It is worth noting that MNDWI showed a warming effect similar to previous studies in part of these areas [50], which may be due to the increase in water temperature

caused by heat-absorbing substances from industries [51]. Specifically, LCZ 2 was the most effective in reducing LST during the summer and transition seasons of Wuhan, probably because LCZ 2 was mainly distributed in "high-high" areas (Figures 2 and 6), where the background temperature was higher, and the water bodies have stronger cooling effects [52]. The highest LST reductions from MNDWI in the summer, transition seasons, and winter of Nanjing were LCZ G, LCZ G, and LCZ 3, respectively, and MNDWI from LCZ G in Shanghai had the strongest cooling effect in all seasons. The cooling effect of MNDWI was stronger in compact building types (LCZ 1–3) than in open building types (LCZ 4–6).

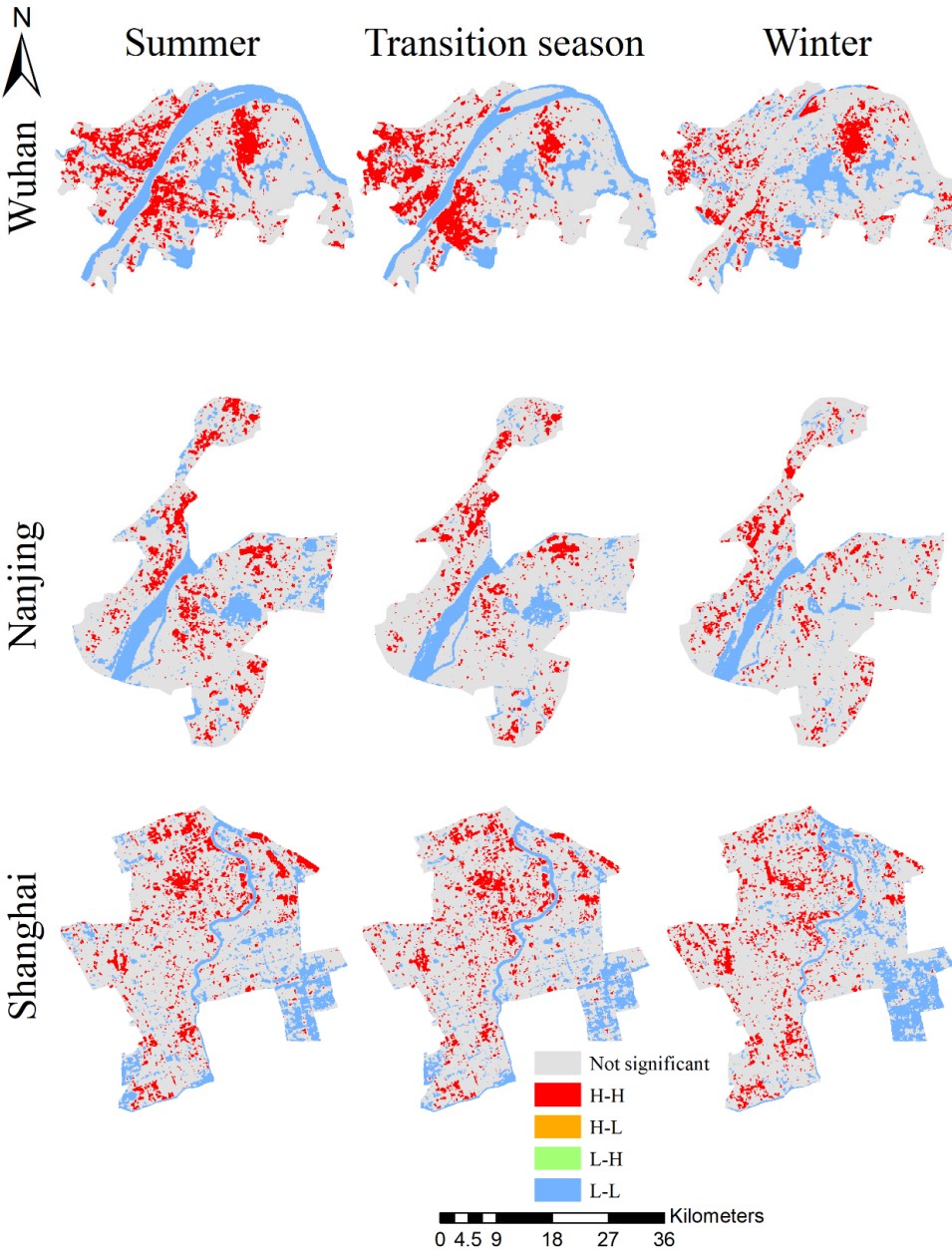

**Figure 6.** Results of local Moran index in three cities and seasons. "H-H","H-L", "L-H" and "L-L" represent "high-high", "high-low", "low-high" and "low-low" type, respectively.

**Table 5.** Diagnosis of GWR model.

| City | Season | MNDWI | NDVI | NDBI |
|------|--------|-------|------|------|
| | Summer | 0.83 | 0.88 | 0.89 |
| Wuhan | Transition season | 0.83 | 0.85 | 0.86 |
| | Winter | 0.75 | 0.75 | 0.78 |
| | Summer | 0.74 | 0.67 | 0.83 |
| Nanjing | Transition season | 0.78 | 0.78 | 0.82 |
| | Winter | 0.57 | 0.56 | 0.54 |
| | Summer | 0.84 | 0.86 | 0.89 |
| Shanghai | Transition season | 0.87 | 0.87 | 0.88 |
| | Winter | 0.79 | 0.80 | 0.81 |

**Table 6.** Change of land-surface temperature with each 0.1 unit increase of spectral indices in different seasons and cities.

| Indices | LCZ | Wuhan Summ | Wuhan Tran | Wuhan Wint | Nanjing Summ | Nanjing Tran | Nanjing Wint | Shanghai Summ | Shanghai Tran | Shanghai Wint |
|---------|-----|------|------|------|------|------|------|------|------|------|
| MNDWI | 1 | −8.98 | −7.78 | −4.03 | −0.12 | −4.57 | −3.88 | −1.53 | −3.25 | −3.97 |
| | 2 | −11.81 | −9.47 | −4.05 | 0.23 | −2.97 | −1.56 | 0.88 | −0.22 | −2.14 |
| | 3 | −7.95 | −7.16 | −4.34 | −5.78 | −6.11 | −5.10 | 1.56 | −0.95 | −1.57 |
| | 4 | −5.29 | −6.71 | −4.90 | 1.33 | −3.01 | −2.65 | 3.20 | 1.68 | −2.25 |
| | 5 | −3.97 | −5.68 | −4.37 | 3.22 | −1.77 | −2.52 | 2.57 | 1.23 | −1.14 |
| | 6 | −2.85 | −4.82 | −4.04 | 7.13 | 1.14 | −1.41 | 5.33 | 2.51 | −0.44 |
| | 8 | −4.32 | −5.88 | −4.85 | 5.14 | 0.68 | −1.72 | 2.86 | 0.55 | −0.98 |
| | 9 | −3.25 | −4.82 | −4.29 | 10.10 | 2.70 | −0.72 | 4.98 | 2.30 | −0.42 |
| | 10 | 2.42 | 2.48 | 0.63 | 3.63 | −2.60 | −2.34 | 1.21 | −0.85 | −1.56 |
| | A | 1.95 | −1.85 | −4.05 | 10.85 | 0.30 | −3.48 | 7.34 | 4.73 | −0.17 |
| | B | −1.18 | −3.99 | −3.95 | 5.50 | −0.08 | −1.92 | 5.38 | 2.22 | −1.21 |
| | C | −6.25 | −6.35 | −3.45 | 3.14 | −1.78 | −2.42 | 5.50 | 2.11 | −1.55 |
| | D | −4.96 | −6.14 | −4.27 | 1.07 | −2.17 | −3.47 | 4.37 | −0.56 | −1.90 |
| | E | 1.12 | −2.73 | −6.33 | −2.97 | −3.44 | −3.89 | 5.99 | 1.84 | −1.15 |
| | F | 0.00 | −4.39 | −4.57 | 2.94 | −1.81 | -2.13 | 2.67 | −0.26 | −1.26 |
| | G | −9.49 | −8.12 | −3.52 | −6.47 | −6.44 | −3.56 | −7.43 | −9.85 | −5.12 |
| NDVI | 1 | 3.04 | 2.69 | 4.65 | −1.43 | 0.63 | 0.86 | −0.87 | 0.33 | 0.92 |
| | 2 | 2.85 | 1.89 | 4.26 | −5.17 | −1.75 | 0.33 | −3.30 | −2.45 | −1.70 |
| | 3 | −1.47 | −2.24 | 1.83 | −0.84 | 0.74 | 3.23 | −4.83 | −2.98 | −0.90 |
| | 4 | −1.15 | 0.35 | 4.85 | −4.08 | −0.83 | 0.33 | −4.26 | −3.03 | −1.54 |
| | 5 | −2.99 | −1.52 | 2.47 | −5.75 | −1.90 | 0.36 | −3.77 | −2.71 | −1.36 |
| | 6 | −3.41 | −1.59 | 1.93 | −9.07 | −4.23 | −0.40 | −5.13 | −3.28 | −1.25 |
| | 8 | −4.86 | −3.54 | 2.74 | −8.26 | −4.71 | −0.63 | −4.36 | −2.22 | −0.82 |
| | 9 | −2.86 | −0.45 | 2.15 | −9.84 | −4.52 | −0.87 | −5.33 | −3.55 | −1.17 |
| | 10 | −6.51 | −7.15 | −4.23 | −5.31 | −0.94 | 0.23 | −2.03 | −0.12 | 0.37 |
| | A | −5.15 | −2.29 | 0.68 | −11.90 | −3.48 | 1.73 | −6.49 | −4.79 | −1.63 |
| | B | −2.42 | −0.42 | 1.63 | −6.82 | −2.22 | 0.84 | −5.02 | −2.91 | −0.51 |
| | C | −0.16 | 0.06 | 0.33 | −0.12 | −0.51 | 1.48 | −0.20 | 0.07 | −0.21 |
| | D | 0.62 | 2.10 | 3.88 | −4.38 | −0.36 | 2.94 | −4.59 | −1.69 | −0.07 |
| | E | −13.55 | −7.40 | −4.19 | −1.47 | 0.97 | 4.94 | −4.53 | −1.48 | 0.78 |
| | F | −1.92 | 2.17 | 5.09 | −4.90 | −0.63 | 1.25 | −4.13 | −1.27 | 0.28 |
| | G | 7.92 | 8.25 | 4.18 | 5.79 | 6.93 | 4.69 | 8.50 | 12.22 | 7.11 |
| NDBI | 1 | 15.09 | 13.37 | 6.76 | 11.76 | 9.05 | 7.74 | 7.25 | 6.26 | 6.57 |
| | 2 | 17.18 | 13.78 | 7.02 | 12.52 | 6.62 | 3.28 | 6.74 | 5.83 | 5.14 |
| | 3 | 18.17 | 12.69 | 6.33 | 17.45 | 13.04 | 7.66 | 9.84 | 7.74 | 4.36 |
| | 4 | 14.73 | 11.98 | 6.15 | 13.43 | 8.87 | 6.50 | 7.72 | 6.13 | 5.68 |
| | 5 | 13.97 | 10.43 | 6.04 | 13.74 | 8.64 | 5.15 | 6.76 | 5.32 | 3.58 |
| | 6 | 17.63 | 12.69 | 5.91 | 14.93 | 8.91 | 3.36 | 8.58 | 5.96 | 3.29 |
| | 8 | 19.78 | 13.56 | 6.87 | 15.51 | 10.19 | 4.37 | 9.51 | 6.52 | 4.20 |
| | 9 | 14.35 | 11.30 | 5.66 | 15.90 | 9.02 | 3.36 | 9.57 | 6.89 | 3.60 |
| | 10 | 18.09 | 13.58 | 4.22 | 14.08 | 8.48 | 5.20 | 8.07 | 5.85 | 3.45 |
| | A | 14.31 | 11.25 | 5.23 | 19.15 | 8.84 | 3.08 | 11.47 | 8.25 | 4.27 |
| | B | 15.65 | 11.95 | 4.95 | 16.39 | 10.26 | 4.41 | 9.85 | 6.68 | 4.13 |
| | C | 18.77 | 16.41 | 5.39 | 14.49 | 10.55 | 5.41 | 9.47 | 7.07 | 3.68 |
| | D | 14.81 | 14.35 | 6.72 | 15.78 | 11.30 | 5.40 | 9.14 | 5.61 | 3.92 |
| | E | 26.43 | 11.88 | 8.81 | 21.13 | 15.22 | 8.02 | 8.74 | 3.74 | 2.13 |
| | F | 13.06 | 17.40 | 6.81 | 14.73 | 10.14 | 5.05 | 8.07 | 5.49 | 2.84 |
| | G | 8.71 | 17.90 | 5.04 | 16.72 | 16.92 | 8.56 | 15.58 | 20.26 | 10.74 |

Note: Summ, Tran, and Wint refer to summer, transition season, and winter, respectively.

NDVI could well reflect vegetation cover, and vegetation reduces LST through transpiration and its own shading [12]. The cooling effect of NDVI in different LCZs also showed seasonal and inter-urban differences. In general, with the decrease of temperature, the cooling effect of each LCZ gradually diminished in summer and transition seasons. In most LCZs of summer and transition seasons, NDVI could reduce LST well, while NDVI showed a warming effect in winter. The cooling effect of open building types was stronger than that of compact building types, which indicated that less green space coverage did not reduce LST well in compact building types. The cooling effect of LCZ A was stronger than that of

LCZ B and LCZ C. There were differences in the effect of NDVI on LST among different cities. In summer and transitional seasons, NDVI had the best cooling effect in LCZ E of Wuhan, while NDVI could achieve the strongest cooling effect in LCZ A of Nanjing and Shanghai.

NDBI reflects the build information. For each 0.1 unit increase in NDBI, all LCZs showed increases of LST in all three seasons and cities. The warming effect of NDBI was stronger in summer and transition seasons than in winter, which may be due to lower sunlight intensity in winter [12]. For Wuhan, the strongest warming effects of NDBI in the summer, transition seasons, and winter were found in LCZ E, G and 8, respectively; for Nanjing, NDBI in different seasons had the strongest warming effect in LCZ E, G, and G, respectively; for Shanghai, the strongest warming effects in different seasons were found in LCZ G.

## 4. Discussion

### 4.1. Thermal Contribution of LCZ and the Effect of SUHII

The results of SUHII showed that there were variations in SUHII amplitude between seasons, and SUHII amplitudes were greatest in the summer, followed by transitional seasons and winter, which is similar to a study in Brno, Prague, and Novi Sad [53]. Further SUHII results were compared with those of the cold, temperate, and tropical studies [1,41,53,54], where the magnitudes of SUHII were different due to differences in background climate and vegetation differences, but higher SUHII values were still found in LCZ 2, 3, 8, 10, and E; LCZ G and A had the lowest SUHII; and LCZ 9 had a smaller SUHII than the other building types in LCZs. The SUHII of LCZ 1 was lower than LCZ 2 and LCZ 3, which might be due to direct shading between high buildings partially blocking solar radiation and preventing solar-surface interaction [42]. The smaller SUHII of LCZ G in this study compared to LCZ A is inconsistent with previous results from eastern Africa and Shenyang, China [1,32], which might be because this study area (especially Wuhan) contained the Yangtze River, with larger and more water bodies to reduce LST [49]. This study found that when the ratio of LCZ A and LCZ D was low, the SUHII of LCZ B and LCZ D was also negative, which meant that LCZ B and D have cooling effects when the percentage of LCZ A was low. Noteworthy, the SUHII of LCZ A in Nanjing was lower compared to Wuhan and Shanghai, which was because Nanjing, with larger and more aggregated LCZ A in the eastern area, had complete ecosystem structures to perform stronger cooling functions. The SUHII of LCZ G in Wuhan was also lower because of the abundant water resources that created a better cooling effect compared with Nanjing and Shanghai.

### 4.2. Policy Implications of Spectral Indexes in LCZs

The ability of MNDWI to reduce LST better in Wuhan compared to Nanjing and Shanghai suggests that evenly distributed and larger water bodies can reduce LST more extensively and effectively. The cooling effect of MNDWI was better in compact building types than in open building types, suggesting that more water bodies should be deployed in compact building types, such as swimming pools on rooftops. The cooling effect of NDVI was stronger in open building types than in compact types in summer and transition seasons, suggesting that there might be a threshold for the cooling effect of green space on LST and that open building types are more effective in reducing LST by accelerating the implementation of green roofs and vertical greening to enhance urban greenery. In summer, NDVI had the strongest cooling effect in LCZ A, which suggests that the implementation of national parks and ecological conservation should be accelerated to protect LCZ A. The cooling effect of NDVI in LCZ A was significantly better than LCZ B and C, so it is recommended that LCZ B and C should be replanted in high heat risk zones ("high-high" type) to reduce the continuous heat risk. With the increase of NDBI, LST increased in all seasons and LCZs. Although the warming effect of NDBI had differences in LST's increase in different LCZs, protecting LCZ G and advocating the use of high-reflectivity materials in LCZ E and 8 are recommended. In addition, the result of SUHII suggests that planners

should avoid concentrated configurations of LCZs 2, 3, 8, 10, and E, as the clustering of these LCZs might further degrade the thermal environment.

### 4.3. Limitations

Some limitations of this study must be acknowledged. First, we selected LST in similar periods as much as possible in the three cities due to the influence of clouds, but it was not possible to make the acquisition times of LSTs in all cities correspond exactly, which affected the comparison results. Second, the spatial pattern of LSTs during nighttime and the response of spectral indexes to LCZs were not analyzed due to the limitation of data. In the future, the LSTs from MODIS and Landsat 8 should be fused to obtain a higher spatial and temporal resolution to analyze the pattern characteristics of LSTs in each LCZ under the daytime and nighttime of different seasons. Finally, this study only analyzed three large cities in the Yangtze River Economic Belt. The impact of spectral indexes on LST should be analyzed in the future for different urban agglomerations and climate zones.

## 5. Conclusions

UHI is a typical problem in urban environments. For these highly heterogeneous urban environments with complex urban morphology, climate, and functions, the impact of spectral indexes on LST is not sufficient. To overcome this deficiency, we analyzed the relationship between spectral indexes and LST under different LCZs to provide a basis for urban planning and policy formulation. We quantified the contribution of each LCZ to the high-temperature zones and SUHII in different cities and seasons, used the Welch and Games–Howell test to detect whether there were significant differences, analyzed the spatial autocorrelation characteristics of LST based on global and local Moran's indexes, and finally used GWR to separately quantify the relationships between MNDWI, NDVI, NDBI, and LST under different LCZs. The results showed the seasonal variations in MDNWI, NDVI, NDBI, and LST, and the largest variation appeared in Wuhan. The proportion of building-type LCZs in the three cities exceeded 50%, ranked from high to low in Shanghai, Nanjing, and Wuhan, with land cover type accounting for nearly 50% in Wuhan. The contribution and SUHII of LCZ 2, 3, 8, 10, and E to the high-temperature zone were higher, and those of LCZ G and LCZ A to the high-temperature zone were lower compared to other LCZs. There were significant differences between 92% and 98% among LCZs and more in summer and transition seasons than in winter. The result of Moran index showed that LST had spatial aggregation characteristics that weakened with decreasing temperature, with the most substantial aggregation characteristics located in Wuhan. The MNDWI in most of LCZs of all seasons for Wuhan could reduce LST well, while MNDWI only had cooling effects in winter for Nanjing and Shanghai. NDVI in most LCZs could reduce LST during summer and transition seasons, while it showed a warming effect in winter. The cooling effect of NDVI in open building types was stronger than that of compact building types, while the cooling effect of MNDWI showed the opposite pattern. With the increase of NDBI, all LCZs showed warming effects, and the magnitude of LST increase varied in different cities and seasons. More water bodies in compact building types and an amount of greening are recommended. Policies on developing national parks and conserving ecology should be accelerated to protect LCZ A. It is suggested that upgrading LCZ B and C to LCZ A by replanting can effectively reduce the thermal risk. Protection of LCZ G and preference for high-reflectivity materials in LCZ E and 8 is recommended. Careful aggregation of LCZ 2, 3, 8, 10, and E is necessary. These results can provide the basis and recommendations for urban planners and policymakers to improve the urban thermal environment.

**Author Contributions:** Conceptualization, Y.X.; data curation, Y.X., Y.T. and Y.D.; formal analysis, Y.X., Y.T. and C.H.; funding acquisition, Z.Z.; investigation, Y.X. and M.T.; project administration, Z.Z.; software, Y.X., C.P., and Y.D.; supervision, Z.Z.; validation, Y.X., C.H. and M.T.; visualization, Y.X., Y.T., Z.W. and C.P.; writing—original draft, Y.X.; writing—review and editing, Z.W. and Z.Z. All authors have read and agreed to the published version of the manuscript.

**Funding:** This research was funded by the National Natural Foundation of China (31870701), the China Scholarship Council (Grant No. 202206760059), and the Science and Technology Projects of Wuhan Municipal Bureau of Landscape and Forestry, Hubei, China (WHGF2018A11).

**Data Availability Statement:** The data presented in this study are available on request from the first author.

**Acknowledgments:** Thanks to the anonymous reviewers.

**Conflicts of Interest:** The authors have no conflict of interest to declare.

## Appendix A

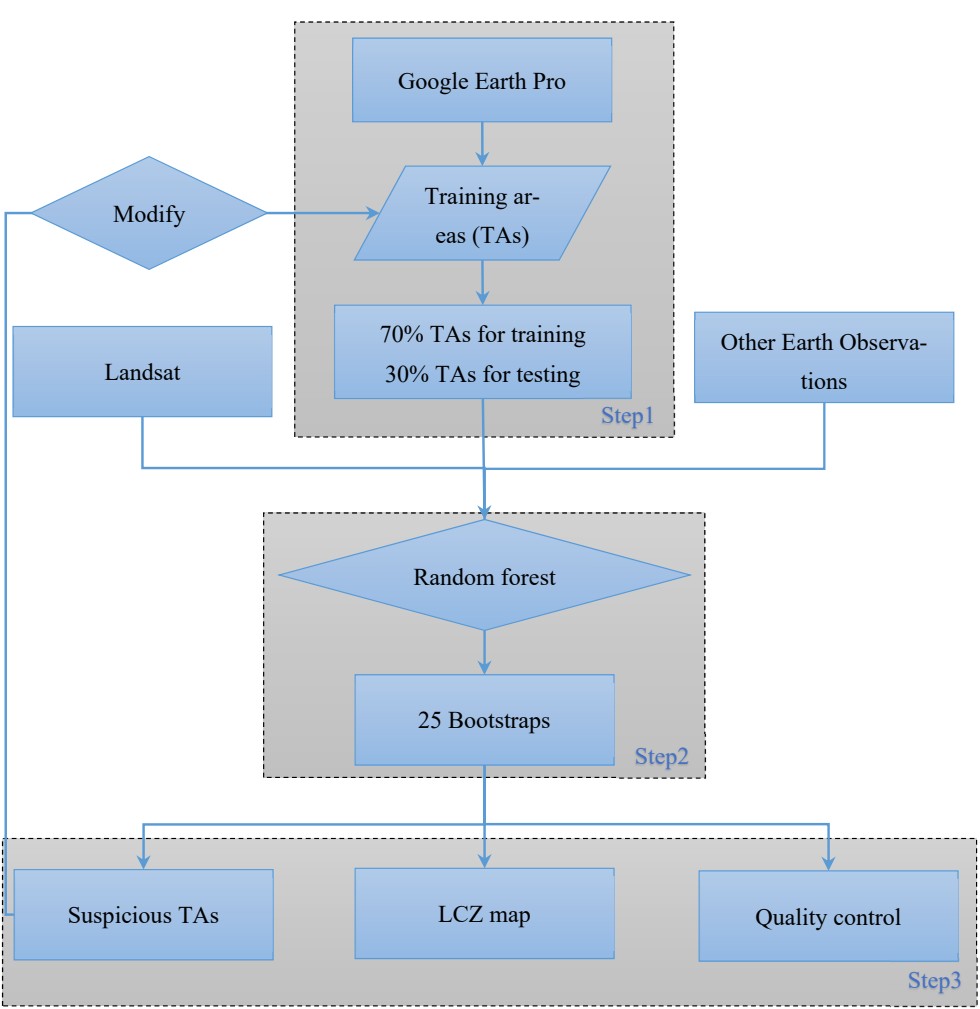

**Figure A1.** Flow chart of local climate zones.

**Table A1.** Descriptions of Landsat 8.

| City | Season | Date | Cloud Cover | Path/Row |
|------|--------|------|-------------|----------|
| Wuhan | Summer | 3 August 2020 | 1.98 | 123/39 |
| | Transition seasons | 13 April 2020 | 3.11 | |
| | | 29 April 2020 | 2.70 | |
| | | 22 October 2020 | 3.82 | |
| | Winter | 25 December 2020 | 0.80 | |

**Table A1.** *Cont.*

| City | Season | Date | Cloud Cover | Path/Row |
|---|---|---|---|---|
| Nanjing | Summer | 11 July 2019<br>12 August 2019<br>1 August 2021 | 18.65<br>12.18<br>12.86 | 120/38 |
| | Transition seasons | 8 April 2020<br>24 April 2020<br>1 October 2020<br>18 November 2020 | 16.82<br>5.05<br>3.39<br>2.69 | |
| | Winter | 20 December 2020<br>6 February 2021<br>22 February 2021 | 0.52<br>12.59<br>19.10 | |
| Shanghai | Summer | 16 August 2020 | 2.17 | 118/38 |
| | Transition season | 12 May 2020 | 15.85 | |
| | Winter | 22 December 2020<br>24 February 2021 | 0.61<br>13.84 | |

**Table A2.** Tests results of normality in different land-surface temperature of local climate zones.

| City | Season | Local Climate Zone | | | | | | | | | | | | | | | |
|---|---|---|---|---|---|---|---|---|---|---|---|---|---|---|---|---|---|
| | | 1 | 2 | 3 | 4 | 5 | 6 | 8 | 9 | 10 | A | B | C | D | E | F | G |
| Wuhan | Summer | | | | | | | | | | | | | | * | | |
| | Transition season | | | | | | | | | | | | | | | | |
| | Winter | | | | | | | | | | | | | | * | | |
| Nanjing | Summer | | | | | | * | | | | | | | | | | |
| | Transition season | | | * | | | | | | | | | | | | | |
| | Winter | | | | | | | | | | | | | | | | |
| Shanghai | Summer | | | | | | | | | | | | | | | | |
| | Transition season | | | | | | | | | | | | | | | | |
| | Winter | | | | | | | | | | | | | | | | |

Note: When the sample size was less than 5000, Shapiro–Wilk test was used; when the sample size was greater than 5000, Kolmogorov–Smirnov test was used. * represents $p > 0.05$ and passes the significance test.

**Table A3.** Levene's tests of land-surface temperature in different local climate zones.

| City | Season | F Statistic | df1 | df2 | Significance |
|---|---|---|---|---|---|
| Wuhan | Summer | 1120.058 | 15 | 95,447 | 0.000 |
| | Transition season | 415.135 | 15 | 95,447 | 0.000 |
| | Winter | 1080.428 | 15 | 95,447 | 0.000 |
| Nanjing | Summer | 114.394 | 15 | 90,073 | 0.000 |
| | Transition season | 166.490 | 15 | 90,073 | 0.000 |
| | Winter | 276.668 | 15 | 90,073 | 0.000 |
| Shanghai | Summer | 414.295 | 15 | 114,552 | 0.000 |
| | Transition season | 493.091 | 15 | 114,552 | 0.000 |
| | Winter | 142.276 | 15 | 114,552 | 0.000 |

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
