# Peer review of "Seasonal Variations of the Relationship between Spectral Indexes and Land Surface Temperature Based on Local Climate Zones: A Study in Three Yangtze River Megacities"

_remotesensing, doi:10.3390/rs15040870_

Round 1
Reviewer 1 Report
It is meaningful to study the effects of blue-green-gray infrastructure on land surface temperatures in megacities. In fact, the biggest innovation was to analyze the relationships between the remotely sensed indices with LST in various LCZs. It's worth exploring. Nevertheless, I cannot recommend this manuscript be published in this form for the following concerns.
1. The title should be changed. Now, the contents cannot support the title.
2. To my knowledge, the closest research to this manuscript is the analysis of the relationship between the remotely sensed indices with LST in different land-cover/land-use types. The authors shouldn’t avoid them.
3. It is not clear that the authors have made any significant new discoveries by analyzing the relationships between the land-use/land-cover indices with LST in various LCZs, especially the ten LCZs of built-up regions. For instance, the authors only explained the negative relationships between MNDWI and LST, but avoid discussing the positive ones in more cases. Moreover, the manuscript almost did not state the similar or different rules in the ten LCZs of built-up regions. In addition, the authors mainly discuss the different relationships between NDVI and LST in non-urban regions, which have been known by us.
4. The differences between remote sensed indices and LSTs have not been analyzed in these three cities. I cannot see the value in choosing three cities.
5. Why did the authors choose three mega-cities in the same climate zone and not in different ones, such as London, Jakarta, Cairo, etc.?
6. Similar descriptions like “LCZ 2, 3, 8, 10, E” should not occur in the abstract. It was not easy to be understood. L21
7. “in transitional season” should be “spring and autumn” in the first time. L25
8. Heat islands have both good and bad effects, and the authors can't just describe the negative effects. L41
9. Please supplement the classification criteria for night light intensiteis in Figure 1. L100
10. Why choose LCZ D as rural region? This is different from many previous studies.
11. The global Moran analysis can be removed.
12. The descriptions of GWR should focus on the research content of this paper.
13. Abbreviations should not be used in headings. L267
14. Have all values passed the significance levels in tables 5 & 6?
15. As the authors said, many scholars have analyzed the difference among LST in different LCZs and the relationships between LSTs and remotely sensed indices. The comparative analysis and discussion between these work and previous ones are very not sufficient.
Author Response
Dear Reviewer,
Your comments have been very helpful. Please see the attachment.

Reviewer 2 Report
The manuscript has good topic selection, detailed method introduction, reasonable structure and proper chart application, so it is a high-quality paper. But let's hope the author thinks about two things:
1. The contrast between Shanghai, Nanjing and Wuhan should be discussed more. The reasons for these differences should be noticed and analyzed by the author.
2. Are the same set of Policy implications of blue-green-grey infrastructure in LCZs reasonable? Should the three cities be distinguished and discussed according to their own characteristics?
Author Response

(The authors gave the same response as above.)

Reviewer 3 Report
Dear authors. I have attached the annotated PDF with minor comments, suggestions, and questions.
You should rewrite the abstract, avoiding the use of undefined terms.
Correct the enumerated citation style following my indications accordingly to the guidelines for authors.
Read and consider my comments. For instance, I suggest the terms reduce and reducing (instead of the terms: mitigate, mitigation, and mitigating).
You should improve the relevance of your good research for international readers by adding more references, examples, and (cross-) comparisons with other similar studies and examples outside China (in the introduction and the discussion) and explain what is novel/original in your study.

Author Response

(The authors gave the same response as above.)

Round 2
Reviewer 1 Report
I am pleased to see these changes of the manuscript. But I still want the descriptions of GWR should focus on the research content of this paper. That means the authors cannot simply introduce this method. The texts are appropriate for any paper. For instance, the author should explain exactly what each variable refers to in GWR.
Author Response
Thank you for your constructive comments. We have modified description of GWR model in section 2.3.7.